

# Identification of SNPs potentially related to immune responses and growth performance in *Litopenaeus vannamei* by RNA-seq analyses

Camilla A. Santos[1,*], Sónia C.S. Andrade[2,*] and Patrícia D. Freitas[1,*]

[1] Departamento de Genética e Evolução, Universidade Federal de São Carlos, São Carlos, São Paulo, Brazil
[2] Departamento de Genética e Biologia Evolutiva, Universidade de São Paulo, São Paulo, São Paulo, Brazil
[*] These authors contributed equally to this work.

## ABSTRACT

*Litopenaeus vannamei* is one of the most important shrimp species for worldwide aquaculture. Despite this, little genomic information is available for this penaeid and other closely related taxonomic crustaceans. Consequently, genes, proteins and their respective polymorphisms are poorly known for these species. In this work, we used the RNA sequencing technology (RNA-seq) in *L. vannamei* shrimp evaluated for growth performance, and exposed to the White Spot Syndrome Virus (WSSV), in order to investigate the presence of Single Nucleotide Polymorphisms (SNPs) within genes related to innate immunity and growth, both features of great interest for aquaculture activity. We analyzed individuals with higher and lower growth rates; and infected (unhealthy) and non-infected (healthy), after exposure to WSSV. Approximately 7,000 SNPs were detected in the samples evaluated for growth, being 3,186 and 3,978 exclusive for individuals with higher and lower growth rates, respectively. In the animals exposed to WSSV we found about 16,300 unique SNPs, in which 9,338 were specific to non-infected shrimp, and 7,008 were exclusive to individuals infected with WSSV and symptomatic. In total, we describe 4,312 unigenes containing SNPs. About 60% of these unigenes returned GO blastX hits for Biological Process, Molecular Function and Cellular Component ontologies. We identified 512 KEGG unique KOs distributed among 275 pathways, elucidating the majority of metabolism roles related to high protein metabolism, growth and immunity. These polymorphisms are all located in coding regions, and certainly can be applied in further studies involving phenotype expression of complex traits, such as growth and immunity. Overall, the set of variants raised herein enriches the genomic databases available for shrimp, given that SNPs originated from nextgen are still rare for this relevant crustacean group, despite their huge potential of use in genomic selection approaches.

Corresponding author
Camilla A. Santos,
camilla.alves@yahoo.com.br

## INTRODUCTION

One of the central challenges in shrimp farming is to avoid economical loss due to animal growth limitations and even population death (*Jung et al., 2013*; *Chen et al., 2015*) commonly caused by pathogens. In an attempt of stop pathogen propagation, the organism goes into a severe oxidative metabolic stress yielding many Reactive Oxygen Species (ROS), which may cause cell and DNA structural damages, leading to apoptosis, and consequently to animal death (*Rewitz et al., 2006*; *Qian et al., 2014*).

Shrimp and other crustaceans do not own a real adaptive immune system as mammals, and are entirely dependent on their innate system, producing a huge variety of immune proteins, including crustacyanin (*Fan et al., 2016*) and hemocyanin (*Zheng et al., 2016*). The antiviral mechanisms and receptors are not well known in crustaceans and some strategies involved in immunity have also been reported, such as nucleic acids that when injected in shrimp organism could be involved in immune gain, as antiviral responses, suggesting a role similar to those assigned to interferons in mammals (*Sadler & Williams, 2008*; *Nehyba, Hrdličková & Bose, 2009*; *Kongton et al., 2011*). Interference-RNAs (RNAi) are also linked to immunity in shrimp, representing a sequence-specific protection against some viral diseases, such as White Spot Syndrome Virus (WSSV) (*Xu, Han & Zhang, 2007*; *Huang & Zhang, 2013*; *Wang & Wang, 2013a*). Mechanisms acting on defense, like phagocytosis, are triggered by the pathogen recognition, binding and encapsulation, leading to cell reorganization and subsequent pathogen destruction through protection strategies, such as acid pH (*Wongpanya et al., 2007*; *Wang et al., 2013b*; *Liu et al., 2014*). Apoptosis are also extremely efficient in preventing virus propagation in the organism, eliminating infected cells and providing some possible advantages in host immunity. Moreover, it is widely known that viral infection and proliferation induces apoptosis in shrimp (*Leu et al., 2013*; *Wang & Wang, 2013a*; *Wang et al., 2014*). In this way, energy that could be being spent in growth is required by shrimp immunity when exposed to pathogens (*Lv et al., 2014*; *Rao et al., 2016*). Moreover, external factors, including temperature and inefficient food supply significantly influences growth performance (*Jindra, Palli & Riddiford, 2013*), leading to a not profitable scenario to aquaculture.

*Litopenaeus vannamei* (Penaeid, Crustacea) is one of the most important shrimp species for worldwide aquaculture, owing a high commercial value aggregated and being an expressive resource for global aquaculture production (*Lu et al., 2018*; *Zhao et al., 2017*). In the meantime, little information is known about the genes and proteins acting on growth and immune responses of this species (*Jung et al., 2013*). Before the lack of a reference shrimp genome, the transcriptomic databases have become essential in identifying new molecular variants, and also in providing data regarding proteins involved in growth and immune system issues (*Rao et al., 2015*; *Sun et al., 2015*). In this way, RNA-seq approaches, based on the next generation sequencing (nextgen), has enabled the gene sequencing high coverage, allowing the large-scale identification of polymorphisms related to economic characteristics in non-model species, such as crustaceans (*Cui et al., 2014*; *Santos, Blanck & Freitas, 2014*; *Yu et al., 2014*; *Santos et al., 2018*). Co-dominant markers, including Single Nucleotide Polymorphisms (SNPs), are highly informative and abundant in genomes

(*Brookes, 1999*; *Vignal et al., 2002*) and have been shown to be efficient in linkage and association studies between molecular and phenotypic data in species such as crabs (*Cui et al., 2014*) and freshwater prawns (*Jin et al., 2013*). However, studies regarding SNPs identified in penaeids by nextgen (*Santos, Blanck & Freitas, 2014*; *Baranski et al., 2014*; *Yu et al., 2014*; *Yu et al., 2017*; *Santos et al., 2018*) and other crustaceans are still incipient until present.

Considering this scenario, the present study was conducted aiming to identify SNPs potentially related to immune responses and growth performance in *L. vannamei*. We analyzed RNA-seq data obtained from individuals with higher and lower growth rates; and infected and non-infected with WSSV, a very aggressive disease in crustaceans that causes enormous losses for shrimp aquaculture industry (*Pradeep et al., 2012*; *Rao et al., 2016*; *Yu et al., 2017*). Our study allowed to identify thousands of specific SNPs in individuals with higher and lower growth rates, and also in healthy and unhealthy shrimps exposed to WSSV. These findings are certainly relevant to be used in further studies considering association of SNPs with immunity and growth performance traits into breeding programs assisted by genomic selection.

## MATERIAL & METHODS

### Biological sampling

The biological samples used in the RNA-seq approach are from muscle and hepatopancreas tissues of aquaculture *L. vannamei* specimens sampled in 2015. We sampled muscle due to its close relation with growth, considering that a faster muscle development may lead to bigger and heavier shrimp, in addition to also be a target for white spot disease (*Jung et al., 2013*; *Shi et al., 2018*). On the other hand, hepatopancreas was sampled, given that most of the immune responses in crustaceans occur in hemocytes, which are produced in this tissue, responsible for storing these cells and many proteins involved in the recognition and elimination of pathogens (*Guo et al., 2013*; *Chen et al., 2015*). Hemolymph samples for the RNA isolation were also collected from animals exposed to WSSV. However, due to the small size of the individuals, and the bad health conditions of the animals with WSSV symptoms, we had no success in the subsequent labor steps.

### *Shrimps evaluated for growth performance (growth group)*

For this approach, we used pleopod samples from Specific Pathogen Free (SPF) *L. vannamei* genetically improved through a selective breeding program conducted for rapid growth performance and good survival rates in a Brazilian shrimp breeding company. We evaluated 20 families and selected individuals from four families with higher and another four with lower growth rates, respectively, according to the quantitative parameter criteria established for the familiar breeding program implemented by the company (data not available). Muscle tissue samples from a total of 48 shrimps (at 45 days of age), belonging to the eight families, were collected, stored in RNA later (Thermo Fisher Scientific, Waltham, MA, USA), and kept in biofreezer ($-80\,°C$) for the RNA isolation.

### Shrimps exposed to WSSV (WSSV group)

For the WSSV-exposure experiments we sampled shrimps from a Brazilian commercial larviculture laboratory. First, SPF commercial Post-Larvae (PLs) were evaluated to white spot virus by qPCR performed according *Silva, Pinheiro & Coimbra (2011)*. The primer pair WSS1011F: 5′-TGGTCCCGTCCTCATCTCAG-3′ and WSS1079R: 5′-GCTGCCTTGCCGGAAATTA-3′ was used in qPCR (*OIE, 2018*) along with Platinum SYBR®Green qPCR Super Mix UDG kit (Thermo Fisher Scientific, Waltham, MA, USA). After confirming absence or presence of the virus, following recommendations proposed by *Pfaffl, Horgan & Dempfle (2002)* and *Ririe, Rasmussen & Wittwer (1997)*, negative PLs to WSSV (WSSV-negative) were transported to a farming land tank and exposed to WSSV. Then, we sampled dozens of unhealthy and healthy shrimps at about two months of age. WSSV symptoms were morphologically detected according *Pradeep et al. (2012)* and *FAO (2018)*. Hemolymph samples from 20 symptomatic and asymptomatic shrimps were collected to perform WSSV-qPCR tests (*Silva, Pinheiro & Coimbra, 2011*; *Pfaffl, Horgan & Dempfle, 2002*; *Ririe, Rasmussen & Wittwer, 1997*), and confirm the presence or absence of the virus. Muscle and hepatopancreas samples were collected for RNA analyses. These samples were stored in RNA later (Thermo Fisher Scientific, Waltham, MA, USA) and maintained at −80 °C. After qPCR confirmation, we selected WSSV positive and negative samples for the RNA essay.

## RNA isolation and cDNA library construction

We isolated total RNA using the Trizol®/chloroform protocol proposed by *Chomczynski & Mackey (1995)*, and checked RNA quality, quantity and integrity in a Q $\mu$bit fluorometer (Thermo Fisher Scientific), a NanoDrop spectrophotometer (Thermo Fisher Scientific), and a BioAnalyser equipment (Agilent Technologies Inc., Santa Clara, CA, USA), respectively. Samples with RNA Integrity Number (RIN) >6,0 were considered proper for later analysis.

We constructed 64 libraries, using a TruSeq RNA Library Preparation V2 kit (Illumina Inc., San Diego, California, USA), for the tissue samples obtained from both growth and WSSV groups. Forty-eight libraries were established for the growth group from (i) 24 muscle samples obtained of animals from the four higher growth families (six shrimps from each family), and (ii) 24 muscle samples of animals from the four lower growth families (six shrimps from each family). Sixteen libraries were established for the WSSV group from (iii) four hepatopancreas samples and (iv) four muscle samples, obtained of four WSSV-positive unhealthy individuals with WSSV symptoms; and (v) four hepatopancreas and (vi) four muscle samples of four WSSV-negative healthy individuals without WSSV symptoms. All animals exposed to WSSV and selected for the library construction were evaluated by qPCR. The unhealthy animals were all positive to WSSV, and the healthy animals were all negative to the virus. We did not find WSSV-positive healthy animals. A small number of WSSV-positive samples from unhealthy shrimps that showed white spot disease symptoms and high quality of RNA was obtained after the trial, limiting the population size for this approach.

## Sequencing, mapping and SNP identification

All cDNA libraries were grouped and sequenced on an Illumina HiSeq 2500 Platform (with 2 × 100 bp paired-end), using a TruSeq SBS V3 kit (Illumina Inc., Thermo Fisher Scientific). The quality of the raw data generated after sequencing was checked in the FastQC software (version 0.10.1) (http://www.bioinformatics.babraham.ac.uk/projects/fastqc/). All reads were filtered for Phred quality (QS) 23 (sequence average) and 30 (sequence edges), and minimum length of 65 bp, using the SeqyClean (v.1.9.9) (https://github.com/ibest/seqyclean). This same software was used to remove contaminant sequences (primers and vectors) listed at the Univec database (https://www.ncbi.nlm.nih.gov/tools/vecscreen/univec/). All reads, available at Sequence Read Archive (SRA-NCBI) under number SRP128934 (BioProject PRJNA428228), were mapped against the reference transcriptome previously *de novo* assembled for *L. vannamei* by *Santos et al. (2018)*.

For the mapping, Bowtie2 v.2.2.6 was applied (*Langmead & Salzberg, 2012*), and the alignment files were used to identify the SNPs. Samtools package (version 1.3) (*Li et al., 2009*) was used to detect SNPs through the mpileup command, using the following parameters: -g -u (calculate the probabilities of the genotypes and generate an uncompressed BCF file), - q20 (minimum mapping quality value for an alignment), -Q20 (minimum quality value of a base), -C50 (minimum mapping quality value used to disregard reads with many mismatches), -A (do not ignore pairs of reads with problems) and -B (do not enable probabilistic realignment, avoiding detection of false SNPs by errors in alignment). SNPs were called through Bcftools (version 1.3) (*Li et al., 2009*). SNPs with variant base quality <30 and sum of the coverage of reads with alternative alleles in the forward and reverse (DP4) strands <10 were excluded from the analyzes to avoid artifacts. Only the variants with a minimum of 50% frequency in each group (higher growth individuals, lower growth individuals, healthy individuals, and unhealthy individuals) were considered for posterior analysis. SNPs with frequency of 50% or more within the animals of each group were selected for analysis in order to increase the reliability of polymorphisms identified, remaining an adequate number of SNPs for further statistics.

## SNP data statistics and functional annotation

Regarding the unigenes with SNPs only, the -max-missing 0.5 command was used in vcftools (version 0.1.14) (*Danecek et al., 2011*) to filter polymorphisms with frequency ≥50%. The number of SNPs, transitions and transversions rate (ts:tv) and multi-allelic sites statistics were computed in bcftools using the –stats command for the SNPs of each of the four groups separately. The Minor Allele Frequency (MAF) was calculated in vcftools with maf 0.05 command, with cutoff value of 5%.

All unigenes, including those with SNPs, were submitted to analysis in the TransDecoder package (http://transdecoder.sourceforge.net/), which was used to identify the contigs candidate coding regions. Meanwhile, Trinotate pipeline (https://trinotate.github.io/) was employed for annotation of the sequences through the following databases: Uniprot (uniref90 + SwissProt) with cut-off value of $1e10^{-5}$, Gene Ontology (GO) (*Ashburner et al., 2000*) for the GO terms Biological Process, Molecular Function and Cellular Component

**Table 1** **Unigenes with SNPs functional annotation.** Overview of SNPs number and functional annotation for the *Litopenaeus vannamei* RNA-seq data obtained from samples evaluated for growth performance and WSSV-exposure.

| | | | | |
|---|---|---|---|---|
| Not redundant unigenes (trimmed for >95% similarity and isoforms). Used in SNPs calling. | | 20,865 | | |
| **Trinotate proteins functional annotation** | | | | |
| BlastX hits unigenes (nr) | | 11,256 (79%) | | |
| **SNPs (frequency ≥50%)** | | | | |
| | **Growth** | | **WSSV-exposed** | |
| | **Higher** | **Lower** | **Healthy** | **Unhealthy** |
| Unigenes with SNPs | 2,300 | | 3,807 | |
| Total unique unigenes with SNPs for growth and WSSV groups | | 4,312 | | |
| | **Higher** | **Lower** | **Healthy** | **Unhealthy** |
| Exclusive SNPs number | 3,186 | 3,978 | 9,338 | 7,008 |
| Total Exclusive SNPs number by group | 7,164 | | 16,346 | |
| Unigenes with GO hits and containing SNPs for growth and WSSV groups | | 2,557 (59%) | | |
| Unigenes with KEGG hits and containing SNPs for growth and WSSV groups | | 844 (20%) | | |
| Total unique KOs number | | 512 | | |
| Total unique KEGG metabolic pathways | | 275 | | |

and KEGG (Kyoto Encyclopedia of Genes and Genomes) (*Kanehisa et al., 2012*), with the identification of KOs (KEGG Ortology) followed by the main participating metabolic pathways.

# RESULTS

## Mapping and SNPs data

The 24 libraries constructed from higher growth samples had 831,930,442 mapped reads, whilst 796,126,113 reads were mapped for the 24 libraries from lower growth samples (96% of the 1,697,401,992 filtered reads). For the WSSV-exposed group, 191,577,540 and 180,169,952 reads were mapped for the muscle tissue of healthy and unhealthy shrimps, respectively. For hepatopancreas, we had 200,513,272 and 183,685,654 reads mapped for healthy and unhealthy individuals, respectively. Therefore, 755,946,418 (96%) reads of samples were mapped against the 786,662,168 reads after SeqyClean filtering. In this way, 2,484,064,160 reads were mapped against 20,865 unigenes previously to SNP calling. Functional annotation against the GO base and identification of KEGG pathways in the Trinotate package were also performed (Table 1).

We identified 3,186 and 3,978 exclusive SNPs (frequency ≥ 50%) (Table S1) for the higher and lower growth samples, respectively, including 123 multi-allelic, gathered in 2,300 unigenes (Table 1). The number of SNPs per unigene ranged from 1 to 18, with 473 (20%) unigenes containing a minimum of five SNPs per unigene (Table S1). Concerning the substitution of bases, the transitions (ts): transversions (tv) rate were of 1.99:1 and 1.97:1 for higher and lower growth samples, respectively (Table 2). MAF was calculated for higher and lower groups highlighting that 98% of all loci in each group shows the second most common allele with frequency greater than 0.5 in the growth population.

**Table 2  Substitution statistics found in *L. vannamei* SNPs.** Transition and transvertions rates observed in the SNPs identified on *Litopenaeus vannamei* transcriptome data obtained for samples evaluated for growth performance and WSSV-exposure.

| Type | Transitions (ts) | | Transvertions (tv) | | | | |
|---|---|---|---|---|---|---|---|
| Polymorphism | CT | GA | AC | AT | GC | GT | Rate (ts:tv) |
| Higher growth | 1,127 (33,7%) | 1,096 (32,8%) | 275 (8,2%) | 390 (11,6%) | 200 (5,9%) | 253 (7,5%) | 1.99 |
| Lower growth | 1,140 (33,6%) | 1,402 (32,7%) | 335 (7,8%) | 516 (12%) | 287 (6,7%) | 303 (7%) | 1.97 |
| WSSV healthy | 3,479 (34,2%) | 3,262 (32%) | 800 (7,8%) | 1,155 (11,3%) | 662 (6,5%) | 815 (8%) | 1.96 |
| WSSV unhealthy | 2,584 (32,6%) | 2,606 (32,9%) | 623 (7,8%) | 988 (12,4%) | 499 (6,3%) | 605 (7,6%) | 1.91 |

Exclusive SNPs (frequency $\geq$ 50%) could also be identified for the WSSV-exposed group, highlighting 9,338 and 7,008 variants for healthy and unhealthy shrimps, respectively (Table 1 and Table S1). We found 434 multi-allelic SNPs distributed in 3,807 unigenes. We identified 1 to 42 SNPs/locus for WSSV-exposed shrimps, with 1,027 (27%) unigenes with five or more SNPs per locus (Table S1). The ts:tv substitution rates were of 1.96:1 and 1.91:1 for healthy and unhealthy samples, respectively (Table 2). Regarding the MAF, healthy and unhealthy animals showed 96% and 97% of loci with the second most frequent allele greater than 5%.

## Uniprot blastX hits on arthropod species

Following the alignment of sequences against Uniprot database, focus was given on genes showing hits for arthropod species and fitness-related functions, such as growth and disease resistance. Among the higher growth shrimp sampling, the SNPs are mostly located in protein genes, such as: actin (ACTY), astakine (ASTA) and ryanodine receptor 44 (RY44). On the other hand, for the lower growth sampling our data showed that the SNPs are mostly in genes of chitinase (CHIT3), chitin deacetylase, hemolymph clottable (CLOT) and cuticle (CU17, CUIA and CUPA3) proteins. For the most frequent exclusive SNPs identified in genes from the WSSV-exposed shrimps we found SNPs in the heat shock stress 22 (HSP22), 60 (CH60) and 67B2 (HS6B), death-associated inhibitor of apoptosis 1 (IAP1), C-type lectin, crustacyanin (CRA2), crustin, hemocyanin (HCYC) and clottable (CLOT) genes only for healthy animals (Table 3). Exclusive SNPs located in relevant protein genes for all shrimp groups analyzed herein were identified. The unigenes with more SNPs per loci and with fitness related functions proteins are listed in Table 3 and Table S1 .

## Functional annotation in GO and analysis of metabolic pathways

After detection of the SNPs, 4,312 unique unigenes containing SNPs (frequency $\geq$ 50%) were established (with hits for any species) and 2,557 returned hits (59%) against the GO database at level 2, considering both samples evaluated for growth performance and exposure to WSSV. The GO results obtained when the groups were analyzed separately are showed in Fig. 1, and also in Supplemental Material for the animals with higher (Table S2) and lower (Table S3) growth performances, and healthy (Table S4) and unhealthy (Table S5) after exposure to WSSV. The main and most frequent ontologies for the shrimps evaluated for growth performance, considering the higher and lower samples, are available in Fig. 1 and Tables S2 and S3. About the WSSV-exposed group, the most relevant ontologies

**Table 3  Main exclusive SNPs identified in unigenes of higher and lower growth performance group and healthy and unhealthy shrimp after WSSV-exposure.** Loci with the highest number of SNPs exclusive to samples evaluated for growth performance and WSSV-exposed that returned blastX hits for arthropod species. The name of the protein and the number of SNPs/locus are also detailed.

| Locus ID | Protein | Exclusive SNP number | |
|---|---|---|---|
| | | **Growth performance** | |
| | | Higher | Lower |
| Locus_30356.0\|BlastHit\|gi\|1729925\|sp\|Q05187.1\|TGMH_TACTR | Hemocyte protein-glutamine gamma-glutamyltransferase | 18 | – |
| Locus_22356.0\|BlastHit\|gi\|728798\|sp\|P41341.1\|ACTY_LIMPO | Actin | 11 | – |
| Locus_30736.0\|BlastHit\|gi\|215273952\|sp\|Q9W5U2.2\|CHIT3_DROME | Chitinase | – | 11 |
| Locus_26398.0 chitin deacetylase 9 precursor [Tribolium castaneum] | Chitin deacetylase | 1 | 10 |
| Locus_28272.0\|BlastHit\|gi\|74897764\|sp\|Q56R10.1\|ASTA_PENMO | Astakine | 10 | – |
| Locus_30850.0\|BlastHit\|gi\|46396031\|sp\|Q9U572.1\|CLOT_PENMO | Clottable | – | 10 |
| Locus_31209.0\|BlastHit\|gi\|33112444\|sp\|Q24498.3\|RY44_DROME | Ryanodine | 10 | – |
| Locus_28234.0\|BlastHit\|gi\|3913391\|sp\|O02387.1\|CU17_BOMMO | Constituent of cuticle | – | 6 |
| Locus_28640.4\|BlastHit\|gi\|3287772\|sp\|P81384.1\|CU1A_HOMAM | Constituent of cuticle | – | 6 |
| Locus_29583.1\|BlastHit\|gi\|5921937\|sp\|P81577.1\|CUPA3_CANPG | Constituent of cuticle | – | 6 |
| Locus_24086.0 antimicrobial peptide type 2 precursor IIc [Pandalopsis_japonica] | Antimicrobial peptide | – | 5 |
| Locus_29787.0\|BlastHit\|gi\|59797979\|sp\|Q9W092.1\|CHIT2_DROME | Chitinase | 2 | – |
| | | **WSSV-exposed** | |
| | | Healthy | Unhealthy |
| Locus_30670.0\|BlastHit\|gi\|55977856\|sp\|Q24306.2\|IAP1_DROME | Death-associated inhibitor of apoptosis 1 | 28 | 26 |
| Locus_31211.0\|BlastHit\|gi\|56405335\|sp\|P37276.2\|DYHC_DROME | Dynein | – | 33 |
| Locus_31257.0\|BlastHit\|gi\|152031623\|sp\|P02515.4\|HSP22_DROME | Heat shock 22 | 42 | 5 |
| Locus_31208.0\|BlastHit\|gi\|152031623\|sp\|P02515.4\|HSP22_DROME | Heat shock 22 | 4 | 14 |
| Locus_30588.0\|BlastHit\|gi\|75026464\|sp\|Q9V895.1\|AN32A_DROME | Acidic leucine-rich | 17 | – |
| Locus_31209.0\|BlastHit\|gi\|33112444\|sp\|Q24498.3\|RY44_DROME | Ryanodine | 16 | – |
| Locus_23466.0 antilipopolysaccharide fator isoform 5 [Fenneropenaeus_chinensis] | Anti-lipopolysaccharide factor | – | 8 |
| Locus_21474.0 C-type lectin [Penaeus_monodon] | Lectin | – | 6 |
| Locus_30118.0\|BlastHit\|gi\|117330\|sp\|P80007.1\|CRA2_HOMGA | Crustacyanin | – | 5 |
| Locus_25051.0\|BlastHit\|gi\|3024418\|sp\|P81060.1\|PEN3C_LITVA | Peneidin | – | 4 |
| Locus_30564.1\|BlastHit\|gi\|122797\|sp\|P80096.1\|HCYC_PANIN | Hemocyanin | 4 | 2 |
| Locus_23573.0 C-type lectin 1 [Marsupenaeus_japonicus] | Lectin | 9 | – |
| Locus_30850.0\|BlastHit\|gi\|46396031\|sp\|Q9U572.1\|CLOT_PENMO | Clottable | 4 | – |
| Locus_25051.0\|BlastHit\|gi\|3024356\|sp\|P81057.1\|PEN2A_LITVA | Peneidin | 1 | – |

related to fitness are showed in Tables S4 and S5 Basically, for unhealthy animals the most common Biological Processes identified were the same as those for healthy with chitin binding (GO:0008061) being highlighted as a Molecular Function (Table S5) (Fig. 1). When searching for the protein sequences predicted for *L. vannamei* against the KEGG database, hits were found for 844 (19.5%) unigenes with SNPs, consisting of 512 unique KEGG Orthologies (KO) distributed in 275 pathways mapped in the KEGG database (Table S6) (Fig. 2).

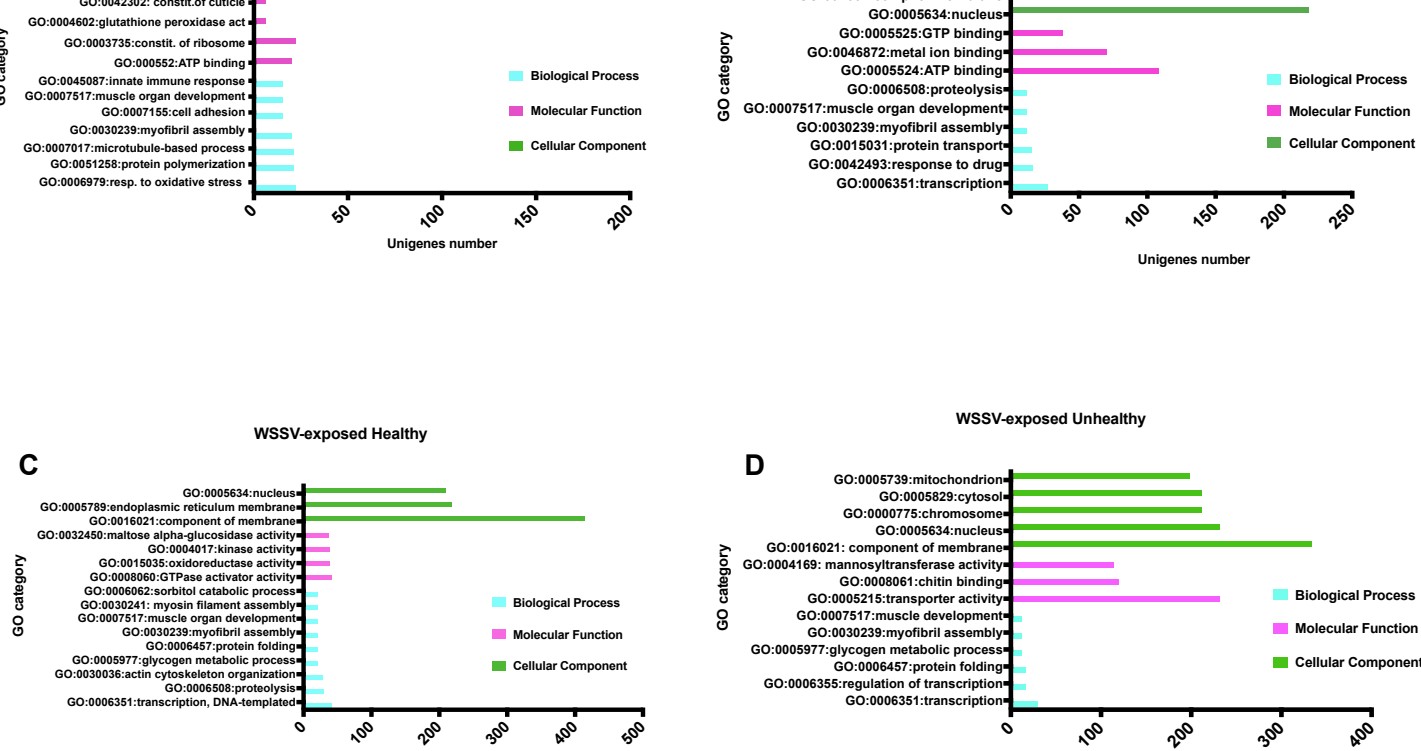

**Figure 1** **Main Gene Ontology blastX hits for unigenes with SNPs.** Distribution of the GO terms found at level 2 for the categories Biological Process, Molecular Function and Cellular Component(*) for (A) higher and (B) lower growth performance group and (C) healthy and (D) unhealthy WSSV-exposed group unigenes. The *x*-axis represents the number of loci that compose each GO term and the *y*-axis the three categories of GO database. Cytoplasm is the most frequent location where cell events take place in all groups (data not shown in chart, but available at Supplemental Information).

## DISCUSSION

### BlastX arthropods performance proteins

#### Shrimps evaluated for growth performance (growth group)

For animals with higher growth performance we found a greater amount of SNPs in genes, such as actin (ACTY), astakine (ASTA) and ryanodine receptor (RY44) (Table 1 and Table S2). Actin is a classic muscle constituent protein certainly linked to growth. This protein was reported to reach higher mRNA levels in the *L. vannamei* shrimp during intermolt and premolt periods, suggesting higher growth rates in shrimp abdominal muscle in these stages (*Cesar & Yang, 2007*). The ryanodine protein is also an important protein acting mainly in the muscle, and its function within the invertebrate group is assumed to be conserved (*Maryon, Saari & Anderson, 1998*). In *Drosophila melanogaster*, previous reports have shown that when ryanodine receptors are blocked, disruption of muscle contraction occurs (*Littleton & Ganetzky, 2000*; *Sullivan et al., 2000*), and consequent impairment of

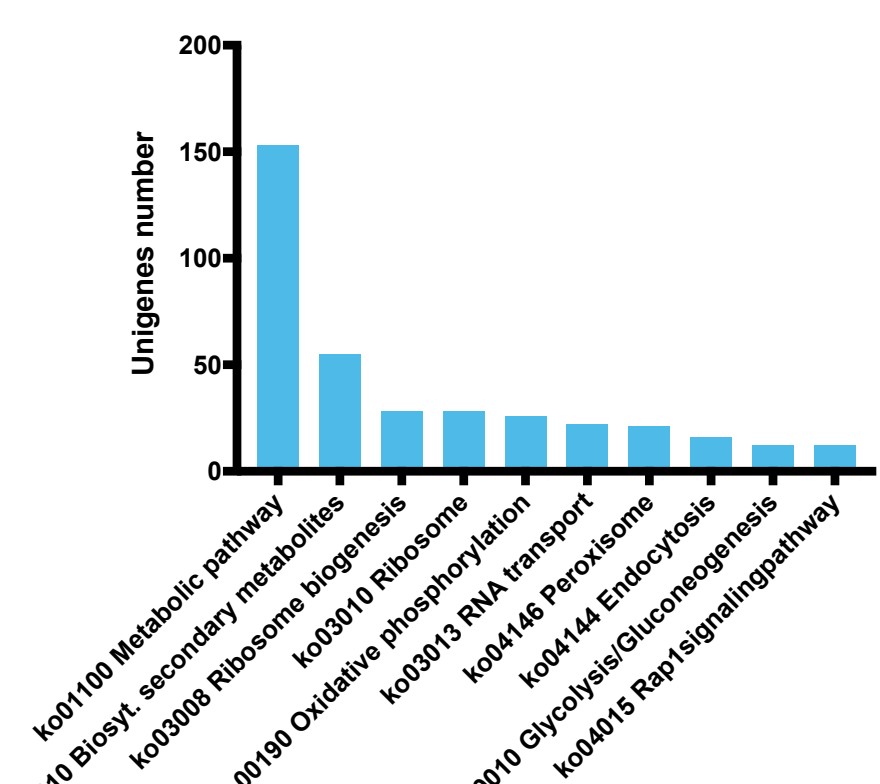

**Unigenes with SNPs and KEGG blastX hits**

**Figure 2 Some of the KEGG pathways with the most KO number.** Pathways mapped in KEGG for RNA-seq data of *Litopenaeus vannamei*. The *x*-axis represents the pathways found and the *y*-axis shows the KO number found in each corresponding pathway.

muscle function. Even though these allelic variants could be observed in unigenes among individuals with higher growth rates only, these data can be useful in further studies considering association of these polymorphisms with growth. Additionally, we also found polymorphisms related to cell maintenance and immunity in higher growth performance shrimps. The astakines are cytokines related to hematopoiesis and antiviral immune responses in crustaceans, participating in the signaling of cell-to-cell responses during immune activation (*Hsiao & Song, 2009*; *Liang et al., 2015*).

Therefore, these higher growth performance shrimps seem to be indirectly selected to immune response as well, though this information should be more explored in further studies regarding such approach. The muscle tissue may be a target for many diseases, such as WSSV, and hemocytes are spread all over the crustacean body, what may explain these polymorphisms in immune genes found here. Another point is that animals used in our study are SPF shrimps evaluated for good survival rates too. Thus, finding allele variants

that may confer some immune advantages in individuals with higher growth and survival rates becomes an interesting strategy in increasing productivity (*Argue et al., 2002*; *Cock et al., 2009*).

For the lower growth animals, we highlight the polymorphisms observed into chitinase and cuticle genes. Chitinase proteins in general are known for causing the rupture of the bounds between chitin molecules and/or breaking the chitin molecules into smaller sugar carbohydrates. The exoskeleton, which is mostly formed by chitin, is digested during the arthropod molt, allowing the body to grow, at the same time that makes the organism highly vulnerable to external adversities, such as diseases (*Proespraiwong, Tassanakajon & Rimphanitchayakit, 2010*). Therefore, a precise balance is required for the animal to grow without being injured. The variants present in CHIT and chitin deacetylase genes, along with the ones in cuticle genes (CU17, CUIA and CUPA3) may be perhaps partially responsible for the lower growth performance of the animals sampled from families with lower growth means. In the case of the exoskeleton takes a longer time to recover after molt, due to chitin formation limitation or chitinase not working efficiently in digesting the cuticle for molt, could be possible reasons for growth deficiency observed in these shrimps (*Zhang et al., 2014*; *Li et al., 2015*). However, the allelic variants observed in these genes and their relation in favoring body growth needs to be investigated in details.

### Shrimps exposed to WSSV (WSSV group)

Proteins related to immune activation and responses were the majority identified in the WSSV-exposed and mostly observed in healthy shrimp. The heat shock proteins (HSPs 22, 60 and 67B2) were the most abundant ones. These proteins often act in conditions of severe stress and are triggered when animals are exposed to environmental disturbances (*Feder & Hofmann, 1999*), showing roles closely linked to proper protein folding and conformation (*Kregel, 2002*; *Tiwari et al., 2015*). Thus, polymorphisms located in these protein genes, in healthy animals, may provide some benefit in allowing and keeping the protein conformation, even before stress conditions caused by diseases, resulting in an easier and more efficient manner to maintain cell homeostasis and activate immune responses.

Other relevant genes that presented SNPs in *L. vannamei* exposed to WSSV and healthy were death-associated inhibitor of apoptosis, hemocyanin, crustin, crustacyanin and lectin protein coding ones. There is a considerable chance that some of the polymorphisms found in this work are related to pathogen recognition inducing phagocytosis, such as lectins (*Song et al., 2010*), with some variants maybe implying some immunological advantages. In arthropods, one of the most common forms of defense against opportunistic microorganisms is endocytosis with consequent encapsulation and destruction of the pathogen (*Wongpanya et al., 2007*). Although, endocytosis may be followed by cell apoptosis, what may not be interesting for the animal as whole, given that when an amount of cells are triggered to death all at the same time, the tissue and later the organism, may be injured, leading even to the animal death (*Liu et al., 2014*). On the other hand, the death-associated inhibitor of apoptosis 1 (IAP1) protein may contribute to reduce the cell damage by acting in apoptosis impairment. As reported in *Wang et al. (2013c)*, when

the expression of LvIAP1 gene was knocked down by dsRNA-mediated gene injection, the level of expression of some genes related to WSSV proteins had their levels increased, emphasizing the role of IAP1 in protection against WSSV. In the case of WSSV-infected shrimp it is suggested that IAP1 protein acts preventing apoptosis of infected cells and avoiding the virus to spread even quicker in the organism, as a result of cell lysis (*Leu et al., 2013*; *Wang et al., 2013c*).

Proteins, such hemocyanin, crustin and crustacyanins have a wide range of action against virus, especially WSSV, acting on animal stress and survival responses (*Fan et al., 2016*), agglutination of the pathogen on hemolymph and cell lysis (*Cheng et al., 2008*; *Shockey et al., 2009*; *Zheng et al., 2016*). The animals used in this work were all obtained from a captive environment and some exposed to a lethal pathogen, what certainly triggers stress responses on their organisms. Thus, the polymorphisms identified in these genes can constitute an important source of information for future studies that evaluate the association of these variants with the immune response.

## GO and KEGG blastX hits

When considering the unigenes, returning blastX for all species, with the most allelic variants that compose the GO terms found for animals evaluated for growth performance, a more active metabolism can be observed in higher growth shrimps, with response to oxidative stress, myofibril assembly, muscle development and innate immune responses as the main events occurring at the cell. In animals with lower growth rates, a more basal metabolism seems to predominate, with transcription leading the BP rank. Nonetheless, response to drugs and proteolysis are also observed as frequent events, maybe illustrating the organism acting in an unexpected way before external disturbances except drugs exposure, such as oxidative stress, high population densities and/or few food supply, what demands higher energy waste and stress, leading to limitations in performance.

Regarding the GO terms identified for the WSSV-exposed shrimps, the healthy individuals showed SNPs mainly in genes related to protein folding and actin cytoskeleton organization. These results may show an intense metabolism with a big amount of energy being required, along with an effective maintenance in protein conformation and function despite the contact with the White Spot Virus, in addition to pathogen endocytosis. In unhealthy animals the events take place more frequently in myofibril assembly and muscle development, together with chitin binding activity. The reason for these GO ontologies may be linked to muscle and exoskeleton cuticle is probably because these are some of the central targets of WSSV in shrimp organism, with the eruption of necrosis white spots in muscle and exoskeleton (*Pradeep et al., 2012*), reflecting a scenario where the animals are showing to be in a constant attempt of recovering themselves from damages caused by the syndrome.

After KEGG blastX hits, 844 unigenes with 512 and 275 unique KOs and pathways, respectively, were identified. In what regards the number of pathways mapped in the KEGG, the value found here was close to other penaied *Illumina* transcriptomes, such as in (*Yu et al., 2014*; *Shi et al., 2018*), in which 240 for *L. vannamei* and 295 for *Fenneropenaeus chinensis* pathways were mapped in KEGG database. These results agreed with the GO

annotation previously described here, that highlighted functions related to an intense protein production and metabolism, central roles in redox signaling before external disturbances, such as diseases, and in dealing with not-recognized particles, such as pathogens. The number of KOs in the pathways, including metabolism, biosynthesis of secondary compounds and ribosome, suggests a high carbohydrate metabolism, for the production of energy, and proteins to supply processes as growth and immune defense. The animals evaluated here were submitted to selection for rapid growth, which characterizes the intense demand for proteins and energy for weight gain. However, the distribution of KOs along the mapped pathways varied widely, as can be seen from the number of pathways with correspondence found (Table S6). The absence of a reference genome and the still small number of transcriptome data in crustacean may restrict a more detailed understanding of the role of some proteins within specific metabolic pathways of more closely related species.

## CONCLUSIONS

The RNA-seq analyses performed herein, using several individuals of *L. vannamei* evaluated for different performance features of interest for aquaculture, enabled the identification of SNPs in important genes related to growth and immunity in crustaceans. In addition, a wide variety of genes with potential fitness-related functions presented several unique SNPs identified for shrimp samples evaluated for (i) higher and (ii) lower growth and (iii) infected (unhealthy) and (iv) non-infected (healthy) after contact with WSSV. Thus, the data generated in this study add relevant information to penaeid transcriptomes, since reports of SNPs detected by nextgen in *L. vannamei* are still rare in the literature. The RNA sequencing provided a wide coverage allied with high resolution of the generated reads, allowing a significant accuracy, reliability and robustness for the SNPs identified here. These polymorphisms may be potentially applied in high-density chips and high-density linkage maps for Genome Wide Association Studies (GWAS) (*Baranski et al., 2014*; *Yu et al., 2014*), providing a base for association analysis between complex traits genotypes and phenotypes. This set of SNPs could be also interesting for allelic-specific expression studies (*Bell & Beck, 2009*), given that all these polymorphisms are located in coding regions and can directly act in phenotype expression. For that, we recommend specific population validations to confirm the presence of such polymorphisms in other *L. vannamei* populations, including outbreed ones, and also related species.

## ACKNOWLEDGEMENTS

The authors thank to Centro de Genômica Funcional do Laboratório de Biotecnologia Animal and Prof. Dr. Luiz Lehmann Coutinho (ESALQ-USP), Piracicaba-SP, Brazil, for the library preparations, sequencing and the computer cluster availability. The authors also thank to Ana Carolina Guerrelhas, Ana Karina Teixeira, Flavio Farias, Karin Kurkjian and João Luis Rocha for supporting the sample collection and providing relevant information related to the fitness performance of the genetically improved shrimp families through a breeding program (private information).

### Funding

This research was supported by the Brazilian agency Fundação de Amparo à Pesquisa do Estado de São Paulo (FAPESP–2012/17322-8 and 2012/13069-6). Conselho Nacional de Desenvolvimento Científico e Tecnológico (CNPq) and Coordenação de Aperfeiçoamento de Pessoal de Nível Superior (CAPES). The funders had no role in study design, data collection and analysis, decision to publish, or preparation of the manuscript.

### Grant Disclosures

The following grant information was disclosed by the authors:
Brazilian agency Fundação de Amparo à Pesquisa do Estado de São Paulo: FAPESP– 2012/17322-8, 2012/13069-6.
Conselho Nacional de Desenvolvimento Científico e Tecnológico (CNPq).
Coordenação de Aperfeiçoamento de Pessoal de Nível Superior (CAPES).

### Competing Interests

The authors declare there are no competing interests.

### Author Contributions

- Camilla A. Santos conceived and designed the experiments, performed the experiments, analyzed the data, contributed reagents/materials/analysis tools, prepared figures and/or tables, authored or reviewed drafts of the paper, approved the final draft.
- Sónia C.S. Andrade performed the experiments, analyzed the data, contributed reagents/materials/analysis tools, authored or reviewed drafts of the paper, approved the final draft.
- Patrícia D. Freitas conceived and designed the experiments, contributed reagents/-materials/analysis tools, authored or reviewed drafts of the paper, approved the final draft.

### DNA Deposition

The following information was supplied regarding the deposition of DNA sequences:
All reads are available at Sequence Read Archive (SRA-NCBI) under number SRP128934 (BioProject PRJNA428228).

### Supplemental Information

Supplemental information for this article can be found online at http://dx.doi.org/10.7717/peerj.5154#supplemental-information.

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
