# Peer review of "Identification of SNPs potentially related to immune responses and growth performance in Litopenaeus vannamei by RNA-seq analyses"

_PeerJ, doi:10.7717/peerj.5154_

## Round 0.1 · original submission · Minor Revisions

Please review the comments from the two reviewers of your manuscript and consider the changes they suggest.

Reviewer 1 ·

Basic reporting

More and precise details of how SNPs were identified are needed, especially, the minimum depth coverage, min number of reads calling SNPs, MAF. It is not clear why max missing 0.5 was used.

The reviewer found this manuscript very lengthy (especially in the results and discussion part), and I strongly suggest that the authors should make it short and concise.

The manuscript could benefit from discussing and citing calculating the allelic imbalances, in addition, the exist/does not exist of SNPs studies in rainbow trout.

Experimental design

A small number of fish was used especially in the disease part. The RIN quality value of the RNA was 1.8-2.2, I am not sure if this a typo or not because this is not an acceptable value.

Validity of the findings

The major problem of this study is that none of the SNPs were validated, at least 10 SNPs or so should be validated by sequencing, or by any other platform, outbred populations should be tested as well. Otherwise, it is hard to make sure these SNPs are high-quality true SNPs.

Reviewer 2 ·

Basic reporting

'no comment'

Experimental design

'no comment'

Validity of the findings

'no comment'

Additional comments

In this manuscript, Camilla A. Santos and colleagues employ RNA-seq analysis to explore the SNPs related to L. vannamei growth performance and innate immune responses. L. vannamei is of great economic importance in worldwide aquaculture. Shrimp growth and immunity are of great interest for aquaculture activity. The results reported in this study enriches the genomic databases available for shrimp. And also, may be potentially applied in high-density chips and high-density linkage maps for Genome Wide Association Studies, association analysis between complex traits genotypes and phenotypes, and allelic-specific expression studies. This study is of interest. However, the following questions should be addressed before recommendation for publication.

Line 54-61, the introduction of shrimp immune system is not precise and comprehensive, e.g. “crustaceans do not own a real immune system”. Shrimp does not have the adaptive immune system but has a powerful innate immune system. Shrimp Toll-like receptors/IMD-NF-kB signaling pathways (PMID: 21827783; PMID: 19232438) have an equivalent role to mammalian TLRs/TNF- NF-kB cascades. Nucleic acid-induced antiviral immunity (PMID: 23773856) and RNAi-, phagocytosis-, and apoptosis-mediated innate immune responses should be extensively described in this part (PMID: 24886688). Otherwise, this part will mislead the readers.

Line 111, “Shrimps exposed to WSSV (WSSV group)”. The sample size is too small, four samples may not representative enough for SNP study (line 117-118). Can the authors explain why they choose muscle and hepatopancreas for cDNA library construction but hemolymph for WSSV-qPCR test? The primers and detailed testing method should be given for testing WSSV.

Line 333, “Shrimps exposed to WSSV (WSSV group)”. The authors report that inhibitor of apoptosis has SNPs in L. vannamei exposed to WSSV. Shrimp IAP1-3 has been shown to play roles in host defense against WSSV (PMID: 23967321), the potential mechanism, such as blocking apoptosis against WSSV infection, should be discussed.

It is amazing that all the polymorphisms are in coding regions. Can the authors give a rational explanation? Are similar findings reported by other publication?

---

## Round 0.2 · accepted · Accept

Your paper was sent out for re-review and has now been accepted for publication.

# Reviewer 2 ·

Basic reporting

no comment

Experimental design

no comment

Validity of the findings

no comment

Additional comments

The authors have improved the work significantly during revision. I recommend its publication now.